# Stapling of Peptides Potentiates the Antibiotic Treatment of *Acinetobacter baumannii* In Vivo

**DOI:** 10.3390/antibiotics11020273

**Published:** 2022-02-19

**Authors:** Gina K. Schouten, Felix M. Paulussen, Oscar P. Kuipers, Wilbert Bitter, Tom N. Grossmann, Peter van Ulsen

**Affiliations:** 1Medical Microbiology and Infection Control (MMI), Amsterdam UMC Location Vumc, De Boelelaan 1108, 1081 HZ Amsterdam, The Netherlands; g.schouten@amsterdamumc.nl (G.K.S.); w.bitter@amsterdamumc.nl (W.B.); 2Department of Chemistry and Pharmaceutical Sciences, Vrije Universiteit Amsterdam, De Boelelaan 1085, 1081 HV Amsterdam, The Netherlands; f.m.paulussen@vu.nl; 3Amsterdam Institute of Molecular and Life Sciences (AIMMS), Vrije Universiteit Amsterdam, De Boelelaan 1085, 1081 HV Amsterdam, The Netherlands; 4Department of Molecular Genetics, Groningen Biomolecular Sciences and Biotechnology Institute, University of Groningen, Nijenborgh 7, 9747 AG Groningen, The Netherlands; o.p.kuipers@rug.nl; 5Department of Molecular Microbiology, Vrije Universiteit Amsterdam, De Boelelaan 1085, 1081 HV Amsterdam, The Netherlands

**Keywords:** stapled antimicrobial peptides, zebrafish larvae infection model, synergy

## Abstract

The rising incidence of multidrug resistance in Gram-negative bacteria underlines the urgency for novel treatment options. One promising new approach is the synergistic combination of antibiotics with antimicrobial peptides. However, the use of such peptides is not straightforward; they are often sensitive to proteolytic degradation, which greatly limits their clinical potential. One approach to increase stability is to apply a hydrocarbon staple to the antimicrobial peptide, thereby fixing them in an α-helical conformation, which renders them less exposed to proteolytic activity. In this work we applied several different hydrocarbon staples to two previously described peptides shown to act on the outer membrane, L6 and L8, and tested their activity in a zebrafish embryo infection model using a clinical isolate of *Acinetobacter baumannii* as a pathogen. We show that the introduction of such a hydrocarbon staple to the peptide L8 improves its in vivo potentiating activity on antibiotic treatment, without increasing its in vivo antimicrobial activity, toxicity or hemolytic activity.

## 1. Introduction

The widespread use of antibiotics has increased the selective pressure on bacteria to develop antibiotic resistance [1]. This rise in resistance is problematic, because only a few novel antibiotics with a new mode-of-action have been approved in the last 30 years [2,3]. Nosocomial infections in particular are prone to develop multidrug resistance, and current therapeutic strategies rely on the activity of some last-resort drugs [4,5,6,7]. Therefore, the World Health Organization published a list of priority pathogens, intended to trigger the development of novel treatments for these infections. The three bacteria with the highest priority are multidrug-resistant Gram-negative bacteria; *Acinetobacter baumannii*, *Pseudomonas aeruginosa* and *Enterobacteriaceae* [8]. The cell envelope of these Gram-negative bacteria, especially the presence of an outer membrane (OM), forms a diffusion barrier for most established antibiotics [9]. Therefore, there is a growing interest in novel therapies and strategies to overcome this barrier. The number of pan-drug-resistant *A. baumannii* strains in particular is growing rapidly and renders it the pathogen with the highest priority for development of new treatments, because this adds to its ability to resist desiccation and to evade host immune defenses [10,11,12].

One of the developing strategies is to use antimicrobial peptides (AMPs) as potentiators of antibiotics that otherwise would not pass the OM in significant amounts to reach their targets. A potentiator compound acts on the OM to allow antibiotics to pass more efficiently and reach their intra-cellular targets. Several AMPs, also known as cationic antimicrobial peptides (CAMPs), are part of the innate immune defence of hosts against a broad range of pathogens [13,14,15]. Although they are usually small and positively charged, they differ in conformation, amphipathicity and hydrophobicity [14,16]. Their commonly attributed mode of action is a non-specific insertion into the membranes of both Gram-positive and Gram-negative bacteria, causing them to be disrupted and become leaky. However, other additional mechanisms of action have been reported [15,17]. Non-specific membrane disruption occurs when the cationic peptides bind negatively charged head groups of lipopolysaccharide (LPS) or phospholipid molecules within the bacterial membranes [17]. Interestingly, CAMPs are less prone to trigger resistance in bacteria than other antimicrobials, especially when administered in combination with other treatments [13,18]. Many established antibiotics that act against Gram-positive bacteria cannot diffuse over the semi-permeable OM of Gram-negative bacteria [9]. Therefore, one promising approach to treat infections by Gram-negative bacteria could be to use a synergistic combination of cell-envelope permeabilizing peptides and established antibiotics [19].

Despite their promising antimicrobial actions, only a few CAMPs, such as colistin and polymyxin B, are used in clinics today. However, many peptides have been described as losing activity under physiological conditions as they are prone to proteolytic degradation, which limits their use to topical applications [18,20,21]. Several attempts have been made to increase the stability of CAMPs, such as D-amino acid substitution, PEGylating, lipidation, macrocyclization and peptide stapling [21,22,23,24,25]. The latter is a technique that introduces an intramolecular side chain-to-side crosslink to increase α-helicity and thereby proteolytic stability [23,25,26]. Different stapled peptides have been described to show potential antimicrobial activity [27,28,29], however only a few have been shown to act against bacteria in vivo and, to our knowledge, stapled peptides have not been described as acting in synergy with antibiotics against bacterial infections in vivo [30,31].

Cecropin A is an AMP produced by the *Cecropia* moth as a humoral immune response against *A. baumannii* and *P. aeruginosa*, which acts by membrane permeabilization and biofilm disruption [32,33]. Melittin is a small amphipathic peptide found in honeybee venom that is known to cause pore-formation in lipid membranes [34,35]. Recently, two cecropin-melittin derived peptides, D6 and D8, have been described to show both antimicrobial activity and potentiating effect when combined with specific antibiotics [30,36]. The peptides were shown to bind LPS and disrupt the outer membranes of Gram-negative bacteria. Interestingly, only the d-peptides reduced bacterial burden when combined with vancomycin in a murine model for abscess formation by *P. aeruginosa* [37]. D-peptides are often not recognized by mammalian proteases, rendering them more stable in vivo. However, manufacturing of d-peptides is costly, which may prevent widespread use of these peptides [18]. In this study, we used an alternative approach to improve peptide stability of the L-form peptides, L6 and L8, namely stapling. We show that by introducing these chemical bridges within the peptide, we can improve the in vivo potentiating activity of peptide L8, without increasing its in vivo antimicrobial activity, toxicity or hemolytic activity.

## 2. Materials and Methods

### 2.1. Bacterial Strains and Growth Conditions

Primary clinical isolates of *Acinetobacter baumannii*, *Escherichia coli* and *Klebsiella pneumoniae*, with kind contributions by Dr. Karin van Dijk, Amsterdam UMC (Table 1), were grown at 37 °C in Luria-Bertani (LB) broth while shaking, or on LB agar plates. Bacteria used for in vitro and in vivo assays were freshly inoculated in LB broth from glycerol stocks for overnight incubation. Overnight cultures were diluted at 1:50 and grown to mid-log phase before use in experiments. 

### 2.2. Solid-Phase Peptide Synthesis

Reagents were used without any additional purification and were purchased from Iris Biotech GmbH (Marktredwitz, Germany), Sigma Aldrich (Darmstadt, Germany), Carl Roth (Karlsruhe, Germany) and Okeanos Biotech (Beijing, China). Peptide sequences were assembled using an automated peptide synthesizer (Syro I, MultiSynTech GmbH, Witten, Germany). The linear peptides L6 (RRLFRRILRWL) and L8 (KRIVQRIKKWLR) were originally designed by Torcato et al. [36] and Qian Li et al. [30], respectively. Peptides were synthesized using Fmoc-based solid-phase peptide synthesis on H-Rink amide ChemMatrix^®^ resin (Sigma-Aldrich, Art. No. 727768, Darmstadt, Germany). Unless stated otherwise, all procedures were performed with 1 mL of solvent or reagent solution per 50 mg resin for all scales (10–100 μmol). The resin was swollen in DMF (dimethylformamide) for 30 min prior to usage. A double coupling protocol of 4 eq. PyBOP (1st 40 min coupling, benzotriazol-1-yl-oxytripyrrolidinophosphonium hexafluorophosphate) and 4 eq. HATU (2nd 40 min coupling, hexafluorophosphate azabenzotriazole tetramethyl uranium) as coupling reagents and DMF as solvent were used. Additionally, the coupling reaction with 4 eq. Fmoc-protected amino acid was supplemented with 4 eq. Oxyma and 8 eq. DIPEA. A capping step was performed using Ac_2_O (acetic anhydride) and DIPEA in NMP(1:1:8, *v/v/v*) for 10 min (2×). Fmoc removal was conducted using 25% (*v/v*) piperidine in DMF (2×) [38].

### 2.3. Olefin Crosslink

For macrocyclization of olefinic non-natural amino acids, ring closing metathesis was performed. Fmoc-protected non-natural olefinic amino acids (Okeanos Tech, Beijing, China) were incorporated in peptide synthesis and treated as natural amino acids. After synthesis, the resin with immobilized peptide was washed and swollen in dichloroethane (DCE) for 15 min. A solution of 4 mg·mL^−1^ Benzylidene-bis(tricyclohexylphosphine)dichlororuthenium (Grubbs Catalyst^TM^ 1st generation) in DCE was added to the resin and reacted at room temperature (RT) for 1.5 h. This procedure was repeated until a sufficient quantity of crosslinked peptide was observed in analytical LC-MS. After metathesis, the resin was washed with DCE, dichloromethane (DCM) and DMF three times. 

### 2.4. Peptide Cleavage and Purification

Cleavage of peptides from resin and removal of sidechain protecting groups was performed simultaneously by incubating the resin with a solution of 94% trifluoroacetic acid (TFA), 2.5% H_2_O and 1% triisopropylsilane (TIPS) twice for 1.5 h. The combined cleavage solution was subsequently reduced under nitrogen flow. Peptides were precipitated using diethyl ether (Et_2_O) at −20 °C and subsequently centrifuged at 4000 rpm for 10 min. The supernatant was decanted and the precipitated peptide lyophilized. Cleaved and lyophilized peptides were dissolved in a solution of 20% acetonitrile (MeCN) in water (+0.1% TFA) and purified by reversed-phase HPLC on an Agilent semi preparative system 1100 (Macherey-Nagel Nucleodur C18 column; 10 mm × 125 mm, 110 Å, 5 µm particle size) using a flow rate of 6 mL·min^−1^ and various gradients of solvent A (H_2_O + 0.1% TFA) and solvent B (MeCN + 0.1% TFA) over 20–40 min. Obtained pure fractions were pooled and lyophilized. 

### 2.5. Characterization

Characterization of peptides was performed by analytical reverse-phase HPLC (1260 Infinity, Agilent Technology; flow rate of 1 mL·min^−1^, A: water with 0.1% FA and 0.01% TFA, B: MeCN with 0.1% FA and 0.01% TFA; Agilent Eclipse XDB-C18 column, 4.6 mm × 150 mm, 5 µm particle size) using a 30 min gradient (5–95% B) coupled to a mass spectrometer (6120 Quadrupole LC/MS, Agilent Technology, Santa Clara, CA, United States) using electrospray ionization. Analytical HPLC chromatograms at 210 nm and MS spectra, masses and *m/z* ratios are shown in the Appendix A. Quantification of acetylated peptides was performed with a NanoDrop OneC using calculated extinction coefficients (https://pepcalc.com/, accessed on 1 February 2021; ε_W_ = 5690 M^−1^cm^−1^, ε_Y_ = 1280 M^−1^cm^−1^, ε_C_ = 120 M^−1^cm^−1^) for λ = 280 nm.

### 2.6. Circular Dichroism Assay 

Peptides were diluted in aqueous 5 mM sodium phosphate solution (pH 7.5) to a final concentration of 7.5 μM. Measurements were performed using a J-1500 CD spectrometer (Jasco, Easton, MD, United States) and a quartz cuvette (10 mm pathlength, Hellma, Müllheim, Germany) at 20 °C. Spectra were recorded in 3 continuous scans at a scanning speed of 100 nm min^−1^ (1 mdeg sensitivity, 0.5 nm resolution, 1.0 nm bandwidth and 2 s integration time). The spectrum of a buffer blank was subtracted from each measurement and the obtained ellipticity (mdeg) was transformed to mean residue ellipticity (MRE/deg cm^2^ dmol^−1^). CD spectra of L6 and L8 were further measured in the presence of three different naturally occurring forms of LPS; smooth (sm) LPS, rough (ra) LPS and deep rough (rd) LPS by adding peptides (fin. conc. 7.5 μM) to a solution of 100 ng/mL respective LPS in the above-mentioned buffer. CD spectra of L6 and L8 were recorded twice and averaged in the analysis. Helicity values were calculated using the CDNN software tool (spectra range for calc.: 195–260 nm, http://gerald-boehm.de/download/cdnn, accessed on 1 January 2021) developed by G. Böhm [39,40].

### 2.7. Minimum Inhibitory Concentration Assay

Minimum inhibitory concentrations of the peptides and antibiotics were determined according to the protocol described by Wiegand et al. [41]. Briefly, two-fold dilution series of cationic peptides in 50 µL volume of LB broth containing 2% DMSO were made in polypropylene 96-well microtiter plates (Ratiolab, L6018123, Dreieich, Germany). The optical density at 600 nm (OD600) of growing bacterial cultures was followed, and at mid-log-phase (OD600 ~0.4–0.5) cultures were diluted to OD600 0.02 and 50 µL was added to the wells, resulting in a total volume of 100 µl, an OD600 of ~0.01 and a DMSO concentration of 1%. An OD600 of ~0.01 accounted for approximately 3 × 10^8^ colony forming units per mL (CFU/mL) for *A. baumannii*, 1 × 10^7^ CFU/mL for *E. coli* and 5 × 10^7^ CFU/mL for *K. pneumoniae* as determined by plate counting. The effect of the peptides on growth was followed in a microplate reader (Synergy H1 Multi-Mode Reader or Synergy HTX Multi-Mode Reader, Agilent Technologies, Santa Clara, CA, United States) by measuring the OD600 every 15 min for 12 h. The minimum inhibitory concentration (MIC), defined as the lowest concentration of compound at which 90% of bacterial growth was prevented, was determined by non-linear regression analysis in GraphPad Prism 8. 

### 2.8. Checkerboard Synergy Assay

As described by Hsieh et al. [42], two-fold dilution series of cationic peptides were made in polypropylene microtiter plates (Ratiolab, L6018123, Dreieich, Germany) in 50 µL volume of LB broth/2% DMSO. In a similar manner, two-fold dilution series of antibiotics were made in polystyrene microtiter plates (Costar, REF3779, Corning NY, United States) in 70 µL volume of LB broth. Then, 50 µL of the antibiotic series was transferred to the peptide containing polypropylene plate, resulting in a checkerboard titration of a peptide on the vertical axis and an antibiotic on the horizontal axis. To some checkerboard assays 15 µg/mL isolated smooth (sm) LPS (from *E. coli* O55:B5, Sigma-Aldrich, L2880), rough (Ra) LPS (from *E. coli* EH100, Sigma-Aldrich, L9641) or deep rough (Rd) LPS (from *E. coli* F583, Sigma-Aldrich, L6893) LPS was added to the LB broth. Mid-log phase cultures were diluted to OD600 0.1 and 10 µL culture was added to each well of the microtiter plate, resulting in a total volume of 110 µL and an OD600 of ~0.01 (~3 × 10^8^ CFU/mL), and the OD600 was measured every 15 min for 12 h using the microplate reader. First, the MIC values of the antibiotics alone and the MIC values of the peptides alone were determined as described previously. Similarly, the MIC values of the antibiotics in combination with the different peptides and the MIC values of the peptides in combination with the antibiotics were determined (the fractional inhibitory concentrations (FIC)) [42]. With these MIC values and FIC values, the fractional inhibitory concentration index (FIC_index_) was calculated as follows: (1)$$\mathrm{FICindex}=\frac{\mathrm{FIC}\;\mathrm{antibiotic}}{\mathrm{MIC}\;\mathrm{antibiotic}\;}+\frac{\mathrm{FIC}\;\mathrm{peptide}}{\mathrm{MIC}\;\mathrm{peptide}}$$FIC_index_ ≤ 0.5 was considered synergistic, whereas FIC_index_ ≥ 2 was considered antagonistic. An FIC_index_ between 0.5 and 1 was considered to be an additive effect.

### 2.9. Microinjection of Zebrafish Larvae 

Transparent Casper zebrafish (*Danio rerio*) larvae were infected and treated according to protocols described by Van der Sar et al. and Bernard et al. [43,44]. In short, zebrafish larvae collected from a laboratory-breeding colony kept at 24 °C and a 12:12 h dark/light regime. Larvae were selected based on morphology according to hours post fertilization (hpf) and kept at 28 °C. All protocols followed the international guidelines on the protection of animals used for scientific purposes specified by the EU Directive 2010/63/EU, which allows zebrafish larvae to be used up to the moment of free-living (5–7 dpf). Larvae were dechorionated at 28 hpf and anaesthetized in 0.2% tricaine methanesulfonate (MS222, Fluka A-5040). Larvae were individually infected with 1 nL, containing approximately 100–150 CFU (1–1.5 CFU/mL), of Gram-negative bacteria via microinjection of the caudal vein as described previously [33]. One hour post infection, larvae were treated with cationic (stapled) peptides, antibiotics or a combination of the two via another microinjection of 1 nL in the caudal vein. To correct dilution of the treatment upon injection in the vein (1 nL in 250 nL zebrafish larvae volume), treatments were diluted in 0.1% phenol red (Sigma-Aldrich, p-0290) in phosphate buffered saline (PBS) to 250 times the concentration as tested in vitro. Zebrafish larvae were kept at 30 °C throughout the experiment and survival of the larvae, based on heartbeat, was determined every 24 h for 5 days. 

### 2.10. Hemolysis Assay 

Defibrinated sheep blood was purchased from BioTrading (LOT 21118SG66, Mijdrecht, The Netherlands) and kept at 4 °C. As previously described by Phan et al. [45], red blood cells (RBCs) were pelleted by centrifugation at 600× *g* for 7 min at 4 °C and gently resuspended in phenol red-free Dulbecco’s Modified Eagle Medium (DMEM, Gibco 21063-029, ThermoFisher Scientific, Waltham MA, United States). This was repeated five times until the supernatant appeared colorless. In the meantime, 10-fold dilution series of the peptides at 50 μM, 5 μM and 0.5 μM in 50 µl DMEM, 1% DMSO were made in polypropylene microtiter plates (Ratiolab, L6018123, Dreieich, Germany). Triton-X100 (1%, Sigma-Aldrich T-8787) and DMSO (10%, Sigma-Aldrich D-8418) were used as positive and negative controls, respectively. Polymyxin B (Sigma-Aldrich, p-1411) and polymyxin B nonapeptide (Sigma-Aldrich, p-2076) were used as reference lytic and non-lytic peptides. Then, 50 µL of RBCs was added to each well. The plate was sealed and incubated at 5% CO2 for one hour at 32 °C, after which the plate was centrifuged at 610× *g* for 5 min at room temperature. Next, 70 µL supernatant was removed and transferred to a flat-bottomed polystyrene microtiter plate (Costar, REF3779) and the OD405, as a measure of released hemoglobin, was determined using a microplate reader (Synergy H1 Multi-Mode Reader or Synergy HTX Multi-Mode Reader, BioTek). Percentage of hemolysis was calculated using the following equation: (2)$$\%\;\mathrm{hemolysis}=\frac{\left(\mathrm{absorbance}\;\mathrm{test}\;\mathrm{sample}\right)-(\mathrm{absorbance}\;\mathrm{diluent})}{\left(\mathrm{absorbance}\;\mathrm{positive}\;\mathrm{control}\right)-(\mathrm{absorbance}\;\mathrm{diluent})}$$

### 2.11. Peptide Stability Assay

The stability measurements were performed in normal pooled human serum (PHS). The respective peptide was diluted to a final concentration of 100 μM in the aforementioned human serum (from 10 mM peptide stocks in DMSO). Then, 100 μM Carbamazepine was used as an internal standard; 60 μL of aforementioned peptide/Carbamazepine solution in PHS was incubated for 1h at 37 °C/750 rpm and 20 μL samples were taken after 0 min, 15 min and 60 min. Next, 1.5 μL of TCA was added to the sample to precipitate serum proteins. The samples were subsequently incubated on ice for 10 min and then centrifuged (12,000× *g*, 10 min). Then, 5 μL of supernatant was diluted in 20 μL of H_2_O/MeCN + 0.1% TFA. The sample was subsequently analyzed by LCMS on an Nucleodur C4-column (Macherey Nagel CC125/4 Nucleodur C4 Gravity, 5 µm) using a linear aqueous MeCN gradient containing FA (0.1%, *v/v*) and TFA (0.01%, *v/v*) as ion pair reagent (5–95%, 8 min). Peptides were quantified using the total ion count (TIC) of selected molecular ions. Remaining peptide quantities were calculated by normalizing the peptide peak areas using the internal standard peak areas and subsequently comparing the obtained value with the 0 min measurement. All measurements were performed in triplicate [46,47].

## 3. Results

### 3.1. L8 Acts Synergistically with Antibiotics against Clinical Isolates In Vitro

L6 and L8 have been suggested as peptides that synergistically increase the effect of vancomycin [30]. We first validated the effect of the potentiator peptides L6 and L8 on the growth of three clinical isolates of *E. coli*, *A. baumannii* and *K. pneumoniae* (Table 1), by adding them separately or in combination with vancomycin in checkerboard assays and determining the MIC of the peptides, vancomycin and their combination. In line with the previous results [30], the MIC values of L6 were lower than those of L8 (Appendix A). The MIC values of vancomycin against the clinical isolates varied between species, with the *A. baumannii* clinical isolate being the most sensitive and the *K. pneumoniae* being the least sensitive to this antibiotic. In combination with vancomycin, the MIC values of L6 decreased 2-fold (Appendix A), whereas the MIC values of L8 decreased up to 174-fold (Appendix A). The drop in MIC value of vancomycin was minimal when combined with L6. However, when combined with L8, the decrease in MIC value of vancomycin was more substantial, with fold-changes up to 220 (Appendix A). Next, the various MIC values were used to determine the fractional inhibitory concentration (FIC) index. An FIC_index_ of 0.5 or lower is indicative of synergy between the compounds; an FIC_index_ between 0.5 and 1 is an additive effect and an FIC_index_ above 1 indicates antagonism [42,48]. In accordance with the published data [26], L6 showed little to no in vitro synergistic activity with vancomycin against the Gram-negative isolates of different species tested, whereas L8 showed in vitro synergy with vancomycin against the Gram-negative pathogens tested (Figure 1A). Interestingly, the synergistic effect of L8 on vancomycin was most prominent and most consistent against *A. baumannii* and *K. pneumoniae.*

### 3.2. Interaction of the Peptides with Lipopolysaccharides Abolishes Synergistic Activity and Increases α-Helicity

Li et al. found that the synergistic activity of peptide D6 and D8 is probably the result of an interaction with o-antigen containing LPS [30]. To substantiate this finding, we tested in vitro synergy of L6 and L8 with vancomycin against *A. baumannii* in the presence of three different forms of *E. coli* LPS: smooth (sm) LPS, rough (ra) LPS and deep rough (rd) LPS [49]. We observed that with all LPS samples, in vitro synergy of L8 with vancomycin against *A. baumannii* was abolished (Figure 1B). This indicates that L8 already interacts with deep-rough LPS consisting of a lipid A domain and only inner core sugar residues. In contrast, the potentiator activity of L6 was not reduced upon addition of the three types of LPS. Interestingly, addition of sm-LPS increased the MIC value of L6 by 10-fold (Appendix A). To investigate a potential interaction between L6 or L8 and LPS further, and to identify potential changes in structural conformation of the peptides upon this interaction, circular dichroism (CD) assays were performed analyzing the peptides alone or in the presence of the different forms of LPS. Based on the obtained spectra, the α-helicity of peptides was determined using CDNN (circular dichroism analysis using neural networks) software [39]. In the absence of LPS, L6 and L8 show low α-helicity (15% and 18%, respectively; Figure 1C). Notably, upon addition of each of the three LPS types, both peptides experience a considerable increase in α-helicity ranging from 46% to 70% (Figure 1C). Taken together, these findings indicate that both peptides interact with LPS, but only the activity of L8 appears to be affected by LPS binding.

### 3.3. L8 Exhibits Toxicity in Zebrafish Larvae

A major bottleneck in the development of peptides for therapeutic use is their toxicity [50,51]. To evaluate potentially toxic effects of the peptides, L6 and L8 alone and combined with vancomycin were injected into the caudal vein of zebrafish larvae to follow the survival of injected larvae. We observed that all larvae injected with a combination of vancomycin and L6 survived (Appendix A). However, some toxicity was observed for the combination of L8 (3.125 µM) and vancomycin (2.5 µM), with only 67% survival of the injected larvae after five days post injection (Appendix A). Vancomycin and L8 alone also caused some adverse effects in the zebrafish larvae (Appendix A). Since the interactions with LPS indicated that the peptides have an affinity for membranes that could have caused the toxic effects, an in vitro RBC hemolysis assay was performed to assess potentially disadvantageous hemolytic properties of the peptides. Peptide concentrations were adjusted to cover the range of in vivo experiments (0.5 µM, 5 µM and 50 µM). Polymyxin B, a toxic AMP that is described as disrupting the membrane of Gram-negative bacteria, was used as a positive control for hemolytic activity [52,53]. Since polymyxin B nonapeptide, the deacylated derivative of polymyxin B, is described as being less toxic than its parental peptide [52], this peptide was used as a negative control for the hemolytic activity of the peptides. L8 did not cause hemolysis in the concentration range tested, whereas L6 caused up to 50% hemolysis at 50 µM (Appendix A). Apparently, the toxicity of L8 is not caused by properties of the peptides that lead to hemolysis. 

### 3.4. Peptide L6 Acts Additively with Vancomycin against A. baumannii In Vivo

Next, the activities of the peptides alone and in combination with vancomycin were tested in vivo with a zebrafish larvae infection model. Since the in vitro checkerboard assays showed the most prominent effects of the combinations of L6 or L8 with vancomycin against *A. baumannii*, this bacterium was chosen as the model pathogen for the in vivo experiments. Zebrafish larvae were infected with *A. baumannii* 1757 and treated with either peptide, vancomycin or with a combination of the peptide and vancomycin via caudal vein injections. Upon the injections, survival of the zebrafish was followed over a period of five days, and the survival percentages were used to determine the FIC_index_. Infection of larvae with 100–150 CFU *A. baumannii* 1757 without treatment resulted in 13% survival and this was reached at 3 days post infection (Figure 1D). Treatment with 2.5 µM vancomycin alone did not change the survival rate, indicating that the antibiotic was not active against *A. baumannii* 1757 at that concentration. The combination treatment of *A. baumannii*-infected zebrafish larvae with vancomycin and L6 resulted in an increased survival (36% for the combination versus 19% survival when treated with vancomycin and 4% survival when treated with L6) (Appendix A and Figure 1D). The increased survival corresponded to an FIC_index_ of 0.65, indicating an additive effect of L6 on antimicrobial activity of vancomycin in vivo, which differs markedly from the antagonistic effects of L6 on antimicrobial activity that was observed in vitro (Appendix A and Figure 1B). Combination treatment of *A. baumannii* infected zebrafish larvae with vancomycin and L8 resulted in only 3% survival (Appendix A and Figure 1D). If we used the separate compounds, we observed survival percentages of 19% and 20% for vancomycin treatment and L8 treatment, respectively (Appendix A). This means that L8 in vivo has no synergistic, and in fact an antagonistic, effect on the activity of vancomycin, probably related to the observed toxicity of L8 in zebrafish larvae. 

### 3.5. Design and Characteristics of Stapled Peptides

Due to the low activity of L8 in the in vivo infection assay, we decided to investigate whether the activity of L8 could be improved by modifying the peptide-by-peptide stapling, which can be expected to increase the α-helical conformation. Helical peptides often exhibit enhanced proteolytic stability and envelope penetrating activity compared to linear peptides [35,54,55,56]. In addition, we observed that L6 and L8 have a predominant α-helical structure when mixed with LPS (Figure 1C) and hypothesized that stabilizing the peptides in an α-helical structure would mimic the active state of the peptide. Peptide stamping was implemented by using hydrocarbon crosslinks, which were introduced through the incorporation of non-proteinogenic amino acids bearing reactive alkenyl sidechains at positions i, i+4 or i, i+7 in L6 and L8 (Figure 2A) [57,58]. The staples were formed using Grubbs I catalyst in a ring closing metathesis (Appendix A) [59]. The positions for the introduction of non-proteinogenic amino acids were selected based on a model put forward by Mourtada et al. [31], which suggests that AMPs commonly have a hydrophilic and a hydrophobic side. Residues on the hydrophobic side that were either 4 or 7 amino acids apart were selected for the implementation of non-proteinogenic amino acids, and thus the crosslinks (S1, S2 and S3, see Figure 2A).

We first investigated the α-helical conformation of the stapled peptides as compared to unstapled peptides L6 and L8 using CD spectroscopy, confirming that indeed there were more pronounced minima at 208 and 222 nm, which is indicative of a higher helicity of the stapled peptides (Figure 2B). Both S1 stapled peptides showed the highest degree of helicity in their series, with 28% for L6S1 and 49% for L8S1 (Figure 2B). In general, the unmodified peptides showed rather low helicity (15% and 15%) (Figure 2B). Next, we explored the stability of the stapled peptides as compared to the unstapled peptides. Peptides were diluted in human serum and peptide quantities over time were determined via HPLCU/MS. We observed that the unstapled peptides were least stable in human serum while L6S2 and L8S3 showed the highest stabilities (Figure 2C).

### 3.6. L8S2 Retains In Vitro Synergy with Vancomycin against A. baumannii

We first tested the in vitro antimicrobial activity of the stapled peptides alone and in combination with vancomycin against clinical isolate *A. baumannii* 1757. The MIC values of the stapled L6 peptides were no more than 2-fold different to the unmodified peptide (Appendix A). On the other hand, the MIC values of the stapled L8 peptides showed greater variability, ranging from a 4-fold increase for L8S1 to a 25-fold improved MIC for L8S3 (Appendix A). Moreover, the synergistic activity of peptides upon combination with vancomycin showed large differences. For example, peptide L8S2 retained a synergistic effect, similar to L8 (Figure 2D), whereas the stapled L8S1 and L8S3 lost their potentiating activity (Figure 2D). 

### 3.7. Most Stapled Peptides Show Little to No Adverse Effects

Subsequently, we examined whether the stapling of the peptides introduced any potentially adverse effects for the survival of zebrafish larvae or hemolytic activity. The larvae injected with the stapled L6 peptides all showed similar survival rates, as the larvae injected with unstapled L6 indicated little toxic effects of these peptides. The sole exception was L6S2, which had a slightly toxic effect, as judged from the decreased survival of the larvae (Appendix A). Interestingly, the larvae injected with the stapled L8 peptides all showed higher survival rates over time compared to the larvae injected with unstapled L8 (Appendix A), indicating that stapling reduced their toxicity. None of the stapled L6 or L8 variants caused hemolysis at the concentrations used in the in vivo experiments (Appendix A). However, at the highest concentration of 50 µM, peptides L6S1, L6S2 and L8S3 were significantly more hemolytic to the RBCs than the negative control (polymyxin B nonapeptide). This concentration is much higher than the concentrations at which they act synergistically with vancomycin against *A. baumannii* in vitro. Peptides L6S3, L8S1 and L8S2 did not cause hemolysis, even at high concentrations (50 µM; Appendix A). 

### 3.8. L8S1 Has a Synergistic Effect on In Vivo Antimicrobial Effect of Rifampicin against A. baumannii

Finally, the antimicrobial activity of combinations of vancomycin and stapled peptides was tested. The stapled versions of L6 did not result in increased survival of zebrafish larvae infected with *A. baumannii* when administered in combination with low amounts of vancomycin. In fact, treatment with stapled L6 slightly reduced the survival of the larvae compared to untreated *A. baumannii* infected zebrafish larvae (Figure 3A). In analogy, L8S2 in combination with vancomycin reduced the survival of *A. baumannii*-infected zebrafish larvae (Figure 3A), suggesting that, similarly to the original L8 peptide, this peptide had a toxic effect on the larvae. In contrast, a combination of vancomycin with L8S1 or L8S3 increased the survival of infected zebrafish larvae from 13% of untreated zebrafish larvae to 33% and 36%, respectively (Figure 3A). Notably, L8S3 alone shows a similar antibacterial activity (Appendix A). This is reflected in the FIC_index_ of 1,56 for the combination treatment of vancomycin and L8S3 (Appendix A). L8S1 on the other hand did not show antimicrobial activity in vivo when administered alone, indicating that increased survival originates from a potentiating activity of L8S1 on vancomycin in vivo. The in vivo FIC_index_ for the combination of vancomycin and L8S1 was determined to be 0.85 (Appendix A). To explore the promising activity of L8S1 further, we tested the in vitro and in vivo effects of this stapled peptide on the activity of two other antibiotics, rifampicin and erythromycin. Here, L8S1 acted synergistically on the in vitro antimicrobial activity of rifampicin, whereas the effect of L8S1 on erythromycin and vancomycin was additive (Figure 3B). Moreover, it was shown that L8S1 did not improve the in vivo antimicrobial activity of erythromycin, but it did increase the in vivo antimicrobial activity of rifampicin (Appendix A). In fact, the combination of rifampicin and L8S1 increased the survival of *A. baumannii*-infected zebrafish larvae from 13% of untreated larvae to 57% survival of the larvae that received the combination treatment (Appendix A and Figure 3C). In contrast, treatment of the infected zebrafish larvae with a combination of the linear peptide L8 and rifampicin reduced the survival of the larvae to 2% (Appendix A). Taken together, these data show that the in vivo activity of L8 can be improved by the peptide stapling approach.

## 4. Discussion

With the rising incidence of multidrug resistant Gram-negative bacteria, the urgency for novel treatment options is growing rapidly [1]. Since interaction of fast-acting cationic peptides with bacterial membranes does not involve a specific target, it has been speculated that bacteria are less prone to develop resistance against these molecules, especially when administered in combination with other drugs [13,18]. One promising approach is treatment with synergistic combinations of established antibiotics acting against Gram-positive bacteria and AMPs or CAMPs [19]. However, as peptides are prone to proteolytic degradation and lose activity under physiological conditions, clinical application of AMPs and CAMPs is currently limited to topical use in ointments only [18,20]. Many attempts have been made to increase peptide stability, among which is hydrocarbon stapling [23,25]. With this technique the peptides are fixed in an α-helical or β-sheet conformation; the secondary structure they are likely to attain upon insertion into the OM renders them less vulnerable to proteolytic degradation [35,54,55,56]. Various stapled peptides have been described as potent antimicrobial drugs, but few have been shown to act against bacteria in vivo and, to our knowledge, none have been described as acting in combination with other drugs [27,28,29]. Here, we describe the effect of hydrocarbon stapling on in vivo activity of two previously described linear antimicrobial peptides, L6 and L8 [30]. Importantly, we demonstrate increased potentiator activity of one stapled variant of L8, L8S1, whereas another stapled peptide, L8S3, showed increased in vitro and in vivo antibacterial activity. 

We corroborate the data of Li et al., 2021 [30], who observed in vitro synergy of L8 with vancomycin against a range of Gram-negative pathogens, including multiple clinical isolates of *A. baumannii* and *K. pneumoniae* (Figure 1A). Peptides are commonly described as being sensitive to proteolytic activity [18]. To avoid this, Li et al. reverted D-amino acid-containing versions of L8 and showed in vivo activity of a modified variant called D11 in a murine abscess infection model using *Pseudomonas aeruginosa*. Here, we show that both linear peptides L6 and L8 are indeed degraded in human serum (Figure 2C), so we expected the linear peptides to perform poorly in vivo. Accordingly, we found that treating *A. baumannii* infected zebrafish larvae with the combination of the unmodified L8 peptide and vancomycin or rifampicin via caudal vein injection did not rescue the larvae from infection (Figure 1D). Surprisingly, treatment of the infected zebrafish larvae with L6 and vancomycin did increase survival of the larvae (Figure 1D). Apparently, in vivo synergy cannot always be predicted using in vitro assays. In complex organisms, other factors such as tissue uptake, activation of the immune system or interaction of the peptides with host molecules could affect the activity of the peptides. For example, two artificially synthesized peptides, HSDF-1 and HSDF-2, have been described as acting in synergy with endogenous lysozyme and pleurocidin against Gram-negative bacteria [20]. Another antimicrobial peptide, epinecidin-1, was shown to have immunomodulatory effects in zebrafish larvae [60]. Moreover, a derivative of vancomycin, VanQAmC_10_, was shown to promote autophagy in macrophages, besides disrupting biofilms of *A. baumannii* [61]. In conclusion, synergy of CAMPs with antibiotics is recommended to be assessed for in vivo activity early in the selection process, and the zebrafish larvae infection model appears to be well suited for this purpose.

In an attempt to improve the stability and in vivo activity of both peptides, we applied three different hydrocarbon staples to the peptides, resulting in six stapled variants. They all showed an increased α-helicity in comparison to the linear parental peptides (Figure 2B), which appears desirable for antimicrobial peptides [35,55,56,62]. Importantly, they were more stable when incubated in human serum (Figure 2C). We showed that one of the peptides, L8S3, had increased antimicrobial activity in vivo in the zebrafish larvae infection model, whereas peptide L8S1, not having a high antimicrobial activity itself, had an additive effect on the antimicrobial activity of vancomycin and acted in synergy with rifampicin against *A. baumannii* in vivo (Appendix A and Figure 3A,C). It is possible that L8 has two separate mechanisms of action, one responsible for potentiator activity and one responsible for antimicrobial activity. In view of the results for the other peptides, however, the increased α-helicity, in vitro antimicrobial and potentiator activity, as well as the serum resistance did not appear to be predictors for effectivity in in vivo. In fact, all stapled L6 variants had reduced in vivo synergy with vancomycin when compared to the unstapled peptide (Figure 3A). On the other hand, we observed that stapling of L8 increased in vivo synergy of the peptide with vancomycin for all stapled variants (Figure 3A,C). Tryptophan-rich and arginine-rich peptides are commonly described as exhibiting strong antimicrobial activity [63]. Additionally, the position of these Trp- and Arg-residues affects the antimicrobial properties of peptides. For example, a variant of the antimicrobial cecropin A-melittin hybrid BP100, called W-BP100, was recently described [64], in which addition of a single N-terminal Trp-residue increased its antimicrobial activity significantly. The different sequences of L6 and L8 possibly result in a different distribution of amphiphilicity along the α-helical conformation, as projected in helical wheel predictions (Figure 2A), leading to different interactions with their putative LPS target.

Alternatively, constraining the otherwise flexible peptides might have an adverse effect on initial steps of such interactions depending on that flexibility, which can differ from peptide to peptide [65]. In any case, a certain flexibility is needed to interact with cell envelope molecules, such as lipids, but the peptides also need some rigidity in order to mechanically cause damage to the bacterial membrane [65,66]. Reducing the flexibility of the peptide by stapling may reduce its interaction with LPS but at the same time increase its ability to mechanically damage the bacterial membrane. A certain balance between these two characteristics may be crucial for high synergistic activity when combined with antibiotics. However, the fact that we observe peptides with similar in vitro characteristics, but with different in vivo potentiator or killing effects, suggests that the mechanisms of action of these peptides are not fully understood yet.

## 5. Conclusions

We show here that combinations of stapled peptide L8S1 and vancomycin may provide an attractive novel therapeutic strategy against nosocomial *A. baumannii* infection. Since bacteria are generally less prone to develop resistance against cationic peptides, especially when administered in combination with other drugs [10,15], we expect that the combination of L8S1 and vancomycin or rifampicin would not be prone to develop resistance in Gram-negative bacteria. Although there is still room for improvement of the potentiating activity of L8S1, this is, to our knowledge, the first report on in vivo synergy between stapled peptides and established antibiotics.

## Figures and Tables

**Figure 1 antibiotics-11-00273-f001:**
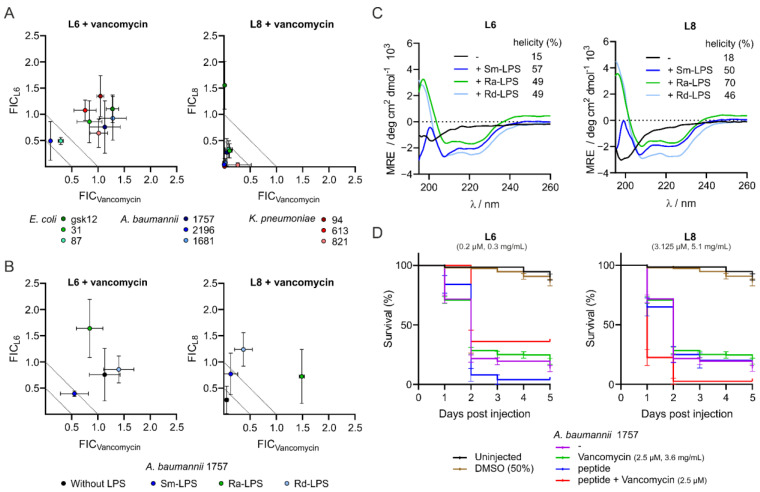
Peptides L6 and L8 are active in vitro against Gram-negative clinical isolates and interact with LPS but have limited activity in vivo. In vitro MIC values of peptides or vancomycin alone and MIC values of peptides combined with vancomycin or vancomycin in combination with peptides against Gram-negative bacteria were determined via checkerboard assays either in medium containing no LPS (**A**) or in medium containing smooth (Sm) LPS, rough (Ra) LPS or deep rough (Rd) LPS (**B**). The FIC values were defined as the ratio of either the MIC value of the peptide in combination with vancomycin over the MIC value of the peptide alone (FIC_L6_ and FIC_L8_) or the ratio of the MIC value of vancomycin combined with peptide over the MIC value of vancomycin alone (FIC_Vancomycin_). The degree of α-helicity of L6 and L8 either in medium containing no LPS, Sm-LPS, Ra-LPS or Rd-LPS was determined with CD spectroscopy. Relative helicity was calculated using circular dichroism analysis using neural networks (CDNN) software (**C**). Zebrafish larvae survival rates after infection with *A. baumannii* 1757 and treatment with peptides, vancomycin or combinations of peptide and vancomycin via caudal vein injection (**D**). The data are presented as mean ± standard deviation from three independent experiments.

**Figure 2 antibiotics-11-00273-f002:**
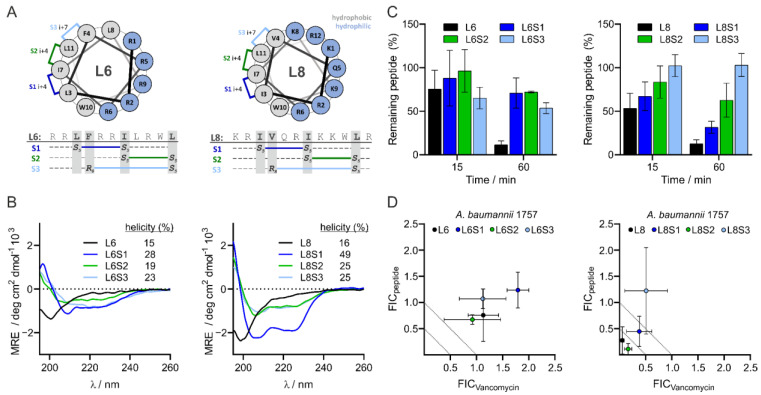
Design and properties of L6 and L8 as well as stapled versions. *Design* and structure of stapled L6 and stapled L8 (**A**). The degree of α-helicity of L6, L8 and the stapled variants of the peptides was determined with CD spectroscopy. Relative helicity was calculated using circular dichroism analysis using neural networks (CDNN) software (**B**). Stability of L6, L8 and the stapled variants of the peptides in human serum were analyzed by LCMS and quantified using total ion count (TIC) of selected molecular ions (**C**). MIC values of the stapled peptides or vancomycin alone and MIC values of the stapled peptides combined with vancomycin or vancomycin in combination with the stapled peptides against Gram-negative bacteria were determined via checkerboard assay. The FIC values were defined as the ratio of either the MIC value of the stapled peptide in combination with vancomycin over the MIC value of the stapled peptide alone (FIC_L6_ and FIC_L8_), or the ratio of the MIC value of vancomycin combined with the stapled peptide over the MIC value of vancomycin alone (FIC_Vancomycin_) (**D**). The data are presented as mean ± standard deviation from three independent experiments.

**Figure 3 antibiotics-11-00273-f003:**
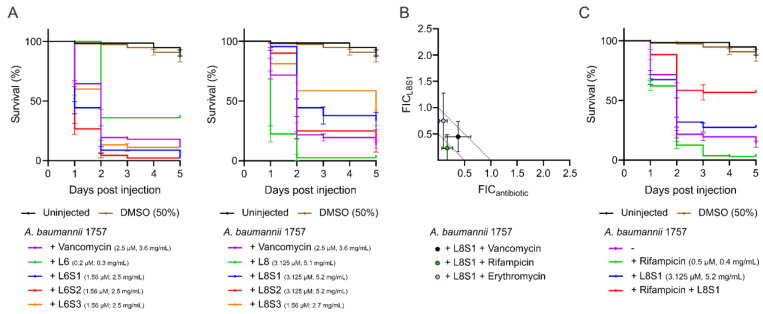
Peptide L8S1 is active in combination with vancomycin or rifampicin against *A. baumannii* in vivo. Zebrafish larvae survival rates were tracked over time following *A. baumannii* infection and treatment with stapled peptides, vancomycin or combinations of stapled peptide and vancomycin via caudal vein injection (**A**). MIC values of L8S1, rifampicin or erythromycin, MIC values of L8S1 combined with either rifampicin or erythromycin and MIC values of rifampicin or erythromycin combined with L8S1 were determined via checkerboard assay. The FIC values were defined as either the ratio of L8S1 in combination with rifampicin or erythromycin over the MIC value of L8S1 alone (FIC_L8S1_), or the ratio of the MIC value of rifampicin or erythromycin combined with L8S1 over the MIC value of rifampicin or erythromycin alone (FIC_antibiotic_) (**B**). Survival rates of *A. baumannii* infected zebrafish larvae were additionally followed upon treatment with peptide L11S1, rifampicin or a combination of L11S1 and rifampicin (**C**). The data are presented as mean ± standard deviation from three independent experiments.

**Table 1 antibiotics-11-00273-t001:** Bacterial strains used in this study.

Species	Strain	Source
*Escherichia coli*	Gsk12	Glaxo SmithKline
*Escherichia coli*	31	Medical Microbiology and Infection Control (MMI), Amsterdam UMC
*Escherichia coli*	87	MMI, Amsterdam UMC
*Acinetobacter baumannii*	1757	MMI, Amsterdam UMC
*Acinetobacter baumannii*	2196	MMI, Amsterdam UMC
*Acinetobacter baumannii*	1681	MMI, Amsterdam UMC
*Klebsiella pneumoniae*	94	MMI, Amsterdam UMC
*Klebsiella pneumoniae*	613	MMI, Amsterdam UMC
*Klebsiella pneumoniae*	821	MMI, Amsterdam UMC

## Data Availability

The data presented in this study is fully disclosed in the paper and in its Appendix A. Further information can be obtained from the corresponding authors.

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
