# Peer review of "Stapling of Peptides Potentiates the Antibiotic Treatment of Acinetobacter baumannii In Vivo"

_antibiotics, 2022, doi:10.3390/antibiotics11020273_

Round 1

Reviewer 1 Report

In this work authors applied several different hydrocarbon staples to L6 and L8 peptides and investigated their activity in Acinetobacter baumannii in vivo infection. It has been shown that the introduction a hydrocarbon staple to the L8, improves its in vivo effect without increasing its toxicity or haemolytic properties. 

The article is very well written. I read it with great interest.

Specific comments. 

  • line 21-22: Is that possible to make this sentence more clear, since it's difficult to understand the difference between "in vivo potentiating activity" and "without increasing its in vivo antimicrobial activity"
  • lines 152-153 - the web link doesn't work 
  • line 154 and line 165: Is that possible to specify approximate doses (concentrations) of bacterial suspensions in CFU/ml
  • line 256: Gram-negative with upper case (here and in other cases of using "Gram-")
  • Please indicate doses also in mg – this is not mandatory, but I think it will be easier for a wider audience – in some important places of the manuscript, where you report toxic and effective in vivo doses; (for example, in figures). 

This manuscript is scientifically sound and suitable for substantiating the hypothesis. 

Author Response

We would like to thank the referee for the favourable review of our manuscript. Please find our responses (printed in bold) to each of your comments below.

  1. line 21-22: Is that possible to make this sentence more clear, since it's difficult to understand the difference between "in vivo potentiating activity" and "without increasing its in vivo antimicrobial activity"

    We agree and have modified the sentence in line 24/25 to make the message more straightforward.

  2. lines 152-153 - the web link doesn't work 

We have checked and corrected (when necessary) the web links provided in the manuscript (line 178 & line 192/193).

  1. line 154 and line 165: Is that possible to specify approximate doses (concentrations) of bacterial suspensions in CFU/ml

To clarify this, we have added the approximate doses of bacterial suspensions used in this study in the materials & methods section of the manuscript. The specific changes can be found in line 201/202/203, 226 and 247.

  1. line 256: Gram-negative with upper case (here and in other cases of using "Gram-")

We corrected this error when occurring in the manuscript.

  1. Please indicate doses also in mg – this is not mandatory, but I think it will be easier for a wider audience – in some important places of the manuscript, where you report toxic and effective in vivo doses; (for example, in figures). 

Upon request, we have indicated the doses of the treatments used in the in vivo experiments both in µM and mg/ml in the respective figures (figure 1D, line 345 and figure 3A and 3C, line 503).

Reviewer 2 Report

Dear Authors

Thank you very much for your manuscript submission. Your work is well-designed but some revisions are needed.

1. The Introduction and Discussion sections should be revised to become much more fruitful. In this regard, please do read and add the following papers to the References section of the manuscript:

Acinetobacter baumannii: An Ancient Commensal with Weapons of a Pathogen. Pathogens. 2021 Mar 24;10(4):387. doi: 10.3390/pathogens10040387. PMID: 33804894; PMCID: PMC8063835.  

Metallo-ß-lactamases: a review. Mol Biol Rep. 2020 Aug;47(8):6281-6294. doi: 10.1007/s11033-020-05651-9. Epub 2020 Jul 11. PMID: 32654052.  

Vancomycin Derivative Inactivates Carbapenem-Resistant Acinetobacter baumannii and Induces Autophagy. ACS Chem Biol. 2020 Apr 17;15(4):884-889. doi: 10.1021/acschembio.0c00091. Epub 2020 Mar 27. PMID: 32195571.

Melittin as a promising anti-protozoan peptide: current knowledge and future prospects. AMB Express. 2021 May 13;11(1):69. doi: 10.1186/s13568-021-01229-1. PMID: 33983454; PMCID: PMC8119515.  

How Insertion of a Single Tryptophan in the N-Terminus of a Cecropin A-Melittin Hybrid Peptide Changes Its Antimicrobial and Biophysical Profile. Membranes (Basel). 2021 Jan 12;11(1):48. doi: 10.3390/membranes11010048. PMID: 33445476; PMCID: PMC7826622.

2. The used protocols should be referred to references.

3. Please use PDB to show 3D structures of the related antimicrobial peptides. An illustrated paper is much more understandable and penetrative.

4. Please show the standard terms relating to used strains in your work (e.g. ATCC etc.)

5. Please add the figures belonging to HPLC technique.

6. The supplementary figures need effective legends.

7. Please do revise the References section. This section needs effective references.

Author Response

We would like to thank the referee for his supportive and useful comments and for reviewing the manuscript. Please find our responses (printed in bold) to each of your comments below.

  1. The Introduction and Discussion sections should be revised to become much more fruitful. In this regard, please do read and add the following papers to the References section of the manuscript:
    1. Acinetobacter baumannii: An Ancient Commensal with Weapons of a Pathogen. Pathogens. 2021 Mar 24;10(4):387. doi: 10.3390/pathogens10040387. PMID: 33804894; PMCID: PMC8063835.
    2. Metallo-ß-lactamases: a review. Mol Biol Rep. 2020 Aug;47(8):6281-6294. doi: 10.1007/s11033-020-05651-9. Epub 2020 Jul 11. PMID: 32654052.  
    3. Vancomycin Derivative Inactivates Carbapenem-Resistant Acinetobacter baumannii and Induces Autophagy. ACS Chem Biol. 2020 Apr 17;15(4):884-889. doi: 10.1021/acschembio.0c00091. Epub 2020 Mar 27. PMID: 32195571.
    4. Melittin as a promising anti-protozoan peptide: current knowledge and future prospects. AMB Express. 2021 May 13;11(1):69. doi: 10.1186/s13568-021-01229-1. PMID: 33983454; PMCID: PMC8119515.  
    5. How Insertion of a Single Tryptophan in the N-Terminus of a Cecropin A-Melittin Hybrid Peptide Changes Its Antimicrobial and Biophysical Profile. Membranes (Basel). 2021 Jan 12;11(1):48. doi: 10.3390/membranes11010048. PMID: 33445476; PMCID: PMC7826622.

In response to the comments and to incorporate the suggested references, we have altered the sections that apply. The specific changes can be found in line  41-44, 95, 97, 98, 553-554 and 574-629. We introduced A. baumannii more extensively in the Introduction section and highlighted the relation of the peptides to cecropin and melittin in the Discussion section of the manuscript.

  1. The used protocols should be referred to references.

We have added the requested references to the Materials and methods section, where appropriate. These were the sections describing Solid-phase Peptide Synthesis, Circular Dichroism Assay, Minimum Inhibitory Concentration assay, Checkerboard Synergy Assay, Microinjection of Zebrafish Larvae, Haemolysis Assay and Peptide Stability Assay (lines 134, 181, 184, 197, 225, 243 and 280).

  1. Please use PDB to show 3D structures of the related antimicrobial peptides. An illustrated paper is much more understandable and penetrative.

Thank you for your suggestion. As we had already illustrated the structure of the stapled peptides in figure 2a we added an illustration of the concept of the ring-closing metathesis and the predicted resulting peptide structures to the supplementary figures. This illustration can be found in supplementary figure 4 and is referred to in line 402 of the main text of the manuscript.  

  1. Please show the standard terms relating to used strains in your work (e.g. ATCC etc.)

The strains used in this study, except for one, are all primary patient isolates that we received from the clinic (MMI, Amsterdam UMC). Typically, such strains are not deposited in strain repositories like ATCC and therefore, lack such codes. However, full clinical data sets are available (patient data and sample data). The numbers indicated in the manuscript are the isolation numbers that were assigned to the different isolates by the clinic. The only bacterial strain that we did not receive from the MMI clinic is the E. coli gks12. This strain is a clinical isolate as well and was obtained from GSK . To clarify that we used only primary clinical strains for the experiments described in this manuscript we have added a note of this in the materials and methods section of the manuscript (line 113). 

  1. Please add the figures belonging to HPLC technique.

The figures belonging to the HPLC testing done to characterize the peptides described in the manuscript can be found in Supplementary figure 1.

  1. The supplementary figures need effective legends.

We agree with the reviewer and have altered the legends and descriptions of the supplementary figures in response.

  1. Please do revise the References section. This section needs effective references.

We have updated and corrected the references in accordance with the suggestions of this reviewer.

Round 2

Reviewer 2 Report

Dear Authors

Thank you very much for your rigorous revision. You added all the recommended papers to the References section. However, you have forgotten to add the following paper, too:

Metallo-ß-lactamases: a review. Mol Biol Rep. 2020 Aug;47(8):6281-6294. doi: 10.1007/s11033-020-05651-9. Epub 2020 Jul 11. PMID: 32654052.